# Development and validation of the Multidimensional Internally Regulated Eating Scale (MIRES)

**Aikaterini Palascha** [1]*, **Ellen van Kleef**[1©], **Emely de Vet**[2©], **Hans C. M. van Trijp**[1©]

**1** Marketing and Consumer Behavior Group, Wageningen University and Research, Wageningen, The Netherlands, **2** Consumption and Healthy Lifestyles Group, Wageningen University and Research, Wageningen, The Netherlands

© These authors contributed equally to this work.

* aikaterini.palascha@wur.nl

**Data Availability Statement:** The data of this research have been uploaded in the OSF Repository (Creation date: 06/03/2020, URL: https://osf.io/w3guh/?view_only= 406839237c9f4bc09ff154d065c17da3).

## Abstract

In this paper, we describe the systematic development and validation of the Multidimensional Internally Regulated Eating Scale (MIRES), a new self-report instrument that quantifies the individual-difference characteristics that together shape the inclination towards eating in response to internal bodily sensations of hunger and satiation (i.e., internally regulated eating style). MIRES is a 21-item scale consisting of seven subscales, which have high internal consistency and adequate to high two-week temporal stability. The MIRES model, as tested in community samples from the UK and US, had a very good fit to the data both at the level of individual subscales, but also as a higher-order formative model. High and significant correlations with measures of intuitive eating and eating competence lent support to the convergent validity of MIRES, while its incremental validity in relation to these measures was also upheld. MIRES as a formative construct, as well as all individual subscales, correlated negatively with eating disorder symptomatology and weight-related measures (e.g., BMI, weight cycling) and positively with adaptive behavioral and psychological outcomes (e.g., proactive coping, body appreciation, life satisfaction), supporting the criterion validity of the scale. This endeavor has resulted in a reliable and valid instrument to be used for the thorough assessment of the features that synthesize the profile of those who tend to regulate their eating internally.

## Introduction

Internally regulated eating (IRE), which can be broadly defined as eating in response to internal, bodily sensations of hunger and satiation, is considered an adaptive way of eating with positive effects on physical, psychological, behavioral, and dietary outcomes [1–6]. IRE has been addressed from various specific theoretical perspectives including, but not limited to, those of intuitive eating [7], eating competence [8], and mindful eating [9]. Palascha et al. [10] recently reviewed these various conceptualizations of IRE to conclude that none of them captures IRE style (i.e., the general inclination towards eating in response to internal/physiological

**Funding:** The authors received no specific funding for this work.

**Competing interests:** The authors have declared that no competing interests exist.

signals of hunger and satiation) comprehensively. The authors conceptualized an integrated model with the key dimensions of IRE style and the relationships between them. The Palascha et al. model suggests that five individual-difference characteristics (detailed below) work as necessary and only jointly sufficient conditions for the manifestation of the IRE style.

Existing measures of IRE, such as the Intuitive Eating Scale 2 (IES-2) [11], the Eating Competence Satter Inventory 2 (ecSI-2) [12], the Mindful Eating Questionnaire (MEQ) [13] and the Mindful Eating Scale (MES) [14] have made impactful contributions, but have failed to capture the full complexity of IRE and the inter-connectedness between the characteristics that define the IRE style. Therefore, there is a need for new measures to assess IRE to its full complexity and potential. The Multidimensional Internally Regulated Eating Scale (MIRES) is proposed to quantify the five individual-difference characteristics that collectively form the IRE style. The present paper reveals the systematic development and validation of the MIRES, a short and easily administered 21-item scale.

In this research we followed a stepwise, theory-based and empirically driven process to develop and validate the MIRES (Fig 1). Next to testing the scale's structure, internal consistency, measurement invariance, and temporal stability, we also examined its content, construct, discriminant, convergent, criterion, and incremental validity. In the next section, we present briefly the conceptual model of the key characteristics of the IRE style, followed by a description of the operationalization of constructs into subscales. For a more complete overview of the conceptual model, including evidence on why each characteristic of IRE style is considered adaptive, see Palascha et al. [10].

## Conceptual definitions and operationalization

Collectively the concept of IRE implies that individuals are sensitive to bodily signals of hunger and satiation, have self-efficacy in using those signals to determine when and how much to eat, trust these bodily signals to guide eating, and have a relaxed and enjoyable relationship with food and eating. *Sensitivity to physiological signals of hunger and satiation* (SH and SS, respectively) is defined as the ability to sense/perceive and interpret the physiological signals that the body generates in response to hunger and satiation. *Self-efficacy in using physiological signals of hunger and satiation* (SEH and SES, respectively) is defined as the perception of ease or difficulty in using physiological signals of hunger and satiation to decide when and how much to eat. *Internal Trust* (IT) refers to the tendency to trust the body's physiological processes for the regulation of eating. *Food Legalizing* (FL) is defined as the tendency to have a relaxed relationship with food and particularly a relaxed attitude towards indulgent food. Finally, *Food Enjoyment* (FE) concerns the tendency to derive pleasure from eating by attending to and appreciating the sensory qualities of the food that is consumed.

IT, FL, and FE are operationalized as uni-dimensional constructs in our model (S1 Fig). Since hunger and satiation are different processes, Sensitivity to hunger signals (SH) and Sensitivity to satiation signals (SS) are operationalized as distinct constructs. The same holds for Self-efficacy in using hunger signals (SEH) and Self-efficacy in using satiation signals (SES). Furthermore, sensitivity and self-efficacy may vary across challenging situations such as when emotional or external cues are salient [15–17]. Therefore, we operationalized each of the constructs mentioned above along three dimensions: under 1. neutral conditions, i.e., when individuals are calm, relaxed, and without much distraction (SH: Neutral, SS: Neutral, SEH: Neutral, SES: Neutral), 2. under emotional prompts, i.e., when negative emotions are salient (SH: Emotional, SS: Emotional, SEH: Emotional, SES: Emotional), and 3. under external prompts, i.e., when external influences, such as a distracting environment, are salient (SH: External, SS: External, SEH: External, SES: External). Since individuals may respond

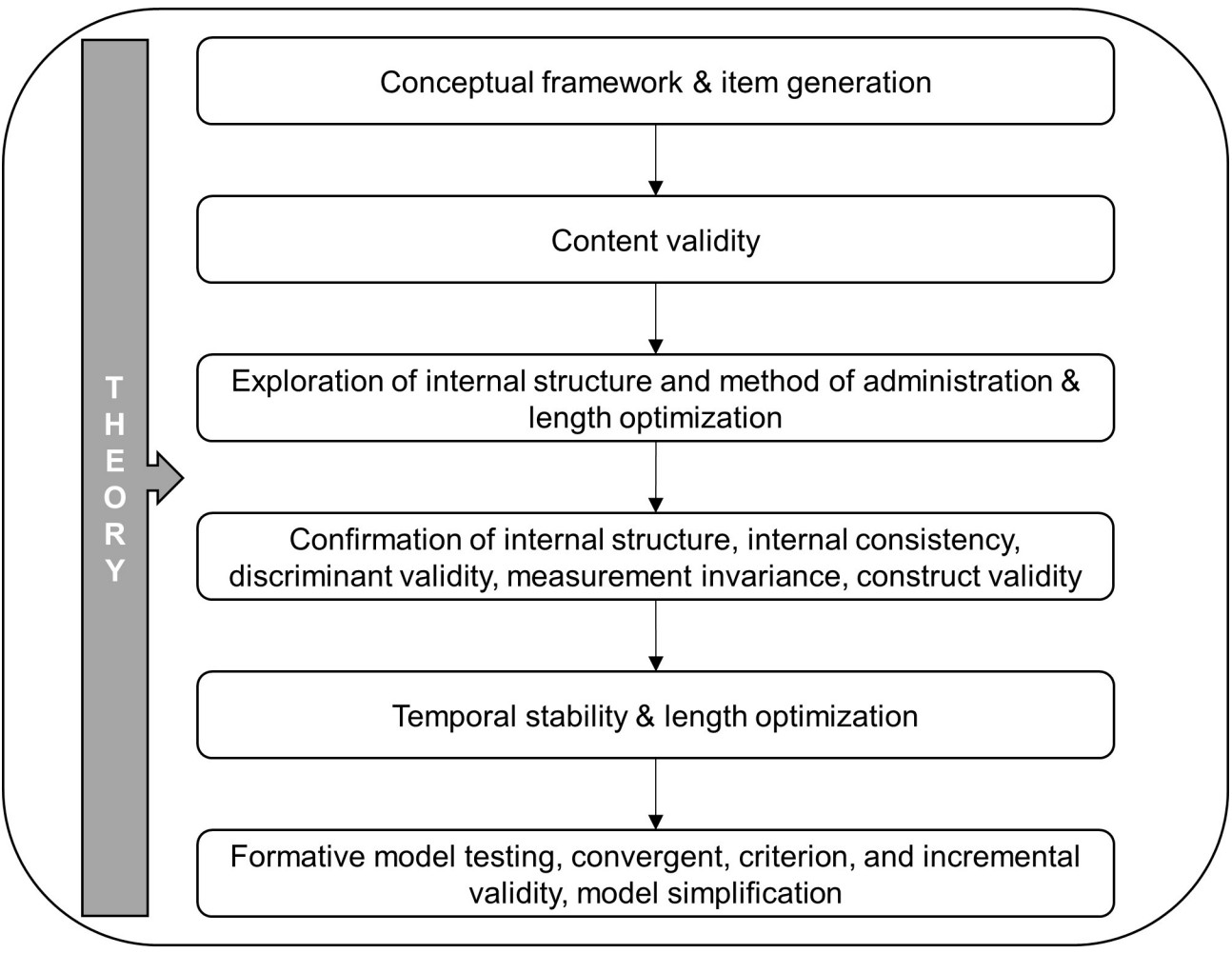

**Fig 1. Steps in the development and validation of MIRES.**

differently to positive and negative emotions, we decided to narrow down to negative emotions. Additionally, high-arousal emotions are assumed to have a universal effect by suppressing eating, while there is more variability in how individuals respond to emotions of moderate arousal [16]. Therefore, only moderate arousal emotional states were selected for the emotional context (i.e., sadness, loneliness, boredom). Regarding the external prompts context, there is a variety of external factors that influence our eating in different ways (e.g., portion sizes, mealtime schedules, eating with others, availability of tasty food, eating in a busy or distracting environment). Given this heterogeneity, we decided to select a single external cue, eating under distraction, because it regards a generic cue that is representative of the process by which several external cues influence eating behavior (i.e., when "noise" from the external environment is salient) and is relevant for both hunger and satiation.

## Model specification

Since the characteristics of the IRE style are not interchangeable—all of them are necessary for the IRE style to manifest—we treated the IRE style as a formative construct. Formative constructs are formed by the combination of their indicators and causality is assumed to flow from the

indicators to the construct [18]. Conversely, a reflective construct exists independently of the indicators that are used to measure it and causality flows from the construct to the indicators. Thus, the IRE style is formed by the totality of its seven defining constructs, while each of these constructs is a reflective one (uni-dimensional or decomposed to measurable sub-dimensions).

## Methods

Through interactive discussions within the author team, we generated a pool of 103 items, which were purported to measure the individual-difference characteristics of the IRE style. Existing measures of intuitive eating [11, 19], eating competence [12], mindful eating [13, 14], and interoceptive awareness [20] were used for inspiration during item generation. Researchers in the field of nutrition and experts evaluated and enriched the content of the initial item pool, which then underwent two rounds of pretesting with college samples. This preliminary work helped us to identify the most appropriate and relevant items for the constructs under study, to sort out the internal structure of the scale, to optimize its length, and to identify the most appropriate method for its administration. Starting from the structure obtained from this preliminary work, we examined the scale's internal consistency, confirmed its internal structure with Confirmatory Factor Analysis (CFA), and tested its two-week temporal stability and several types of validity (i.e., construct, discriminant, convergent, criterion, and incremental) in broad samples of consumers from the UK and US (Table 1). This research was conducted according to the guidelines laid down in the Declaration of Helsinki and complied with the Netherlands Code of Conduct for Research Integrity. Written consent was obtained for all survey participants. Participants who were recruited via market research agencies had previously consented to participate in the panel of the agency. This research was approved by the Social Sciences Ethics Committee of Wageningen University and Research. The data of this project can be found here [21].

### Measures

#### Internally regulated eating

MIRES was administered with 7-point Likert-type response scales (1 = "Completely untrue for me" to 7 = "Completely true for me") (see S1 Appendix for information on administration of

**Table 1. Overview of sample characteristics.**

|  | UK sample (N = 974) | UK sub-sample (N = 213) | US sample (N = 1200) |
|---|---|---|---|
| **Gender** | | | |
| Males | 417 (42.8) | 102 (47.9) | 590 (49.2) |
| Females | 557 (57.2) | 111 (52.1) | 610 (50.8) |
| **Age** | | | |
| 18–24 | 105 (10.8) | 16 (7.5) | 183 (15.3) |
| 25–34 | 174 (17.9) | 27 (12.7) | 253 (21.1) |
| 35–44 | 214 (22.0) | 42 (19.7) | 255 (21.3) |
| 45–54 | 235 (24.1) | 58 (27.2) | 277 (23.1) |
| 55–65 | 246 (25.3) | 70 (32.9) | 232 (19.3) |
| **Education level** | | | |
| Low | 94 (9.7) | 20 (9.4) | 84 (7.0) |
| Middle | 438 (45.0) | 101 (47.4) | 360 (30.0) |
| High | 442 (45.4) | 92 (43.2) | 756 (63.0) |

Values are presented as counts (percentages).

the MIRES). The MIRES items were developed and tested in the English language. An overview of the initial item pool and the adjustments it was subjected to during the scale development and validation process can be found in S2 Appendix.

A necessary condition for identification of formative models is the addition of at least two reflective measures that are caused directly or indirectly by the formative construct [22]. Thus, to achieve identification when testing the complete formative model we also developed six items that were reflective of the higher-order factor IRE style. We use the abbreviation *RI* (Reflective items) to refer to these items in the rest of the paper. Cronbach's alpha for the RI was 0.90 and AVE was 0.61. Uni-dimensionality of the RI factor was supported by the good model fit ($\chi^2$ (9) = 110.68, p < 0.001, CFI = 0.98, TLI = 0.96, RMSEA = 0.10, SRMR = 0.03) and the high factor loadings (0.68–0.85).

## Intuitive eating

We measured intuitive eating to test the convergent and incremental validity of MIRES. The 21-item IES-2 [11] was used to measure the four constructs of intuitive eating, namely, Unconditional Permission to Eat (UPE), Eating for Physical Rather Than Emotional Reasons (EPR), Reliance on Hunger and Satiety Cues (RHSC), and Body Food Choice Congruence (BFCC). Items were administered on a 5-point scale (1 = "Strongly disagree" to 5 = "Strongly agree"). Cronbach's alphas were 0.69 (UPE), 0.87 (EPR), 0.93 (RHSC), and 0.88 (BFCC).

## Eating competence

We also measured eating competence to test the convergent and incremental validity of MIRES. The 16-item Eating Competence Satter Inventory 2.0 (ecSI-2) was used to measure the four constructs of eating competence [12, 23]; Eating Attitudes (EatAtt), Food Acceptance (FoodAccept), Internal Regulation (IntReg), and Contextual Skills (ContSkills). Items were administered on a 5-point scale (1 = "never" and 5 = "always") and responses were used as continuous variables in this study. Cronbach's alphas were 0.88 (EatAtt), 0.75 (FoodAccept), 0.84 (IntReg), and 0.83 (ContSkills).

## Eating disorder symptomatology

The Binge Eating Scale (BES) and the Restrictive Eating Scale (RES) of the Multifactorial Assessment of Eating Disorder Symptoms (MAEDS) [24] were used to assess the frequency of manifesting binge eating and restrictive eating behaviors. Items were administered on a 7-point frequency scale (1 = "Never" to 7 = "Always"). Two items from each subscale were dropped before data collection ("I crave sweets and carbohydrates" because it regards a behavior that is non-specific for binge eating and had a low item-total correlation in the original study; "I am too fat" because it reflects a belief rather than a behavior; "I eat 3 meals a day" because it is the only item with negative item-total correlation and because for some people it may seem as a stringent behavior, while for others as an adaptive one; "I hate to eat" because it was deemed extreme and had a low item-total correlation in the original study). Cronbach's alphas for the adapted scales were 0.91 (BES) and 0.87 (RES). The fit of the RES model was initially unacceptable. Thus, we allowed for correlated error terms between the two items on fasting that have similar wording. BES and RES were measured to assess the criterion and incremental validity of MIRES.

## Proactive coping

The 8-item Proactive Coping Scale (PCS) of the Proactive Coping Inventory, as adapted by Gan et al. [25], was used to measure cognitions and behaviors related to self-regulatory goal

attainment. Items were administered on a 4-point scale (1 = "Not at all true" to 4 = "Completely true"). The PCS model fit was improved by allowing for correlated error terms between the items that refer to dealing with challenges as there is word congruence among them. We further removed the two reverse-scored items after data collection because of low item-total correlations (0.184 and 0.165, respectively). The adapted PCS had a Cronbach's alpha of 0.88. PCS was measured to assess the criterion and incremental validity of MIRES.

### Adaptive eating behaviors

Two adaptive eating behaviors from the Adult Eating Behavior Questionnaire (AEBQ) were assessed [26]. Satiety responsiveness (SR) assesses with four items the tendency to respond to internal satiety signals. Slowness in eating (SE) measures with four items the tendency to consume meals at a slow pace. Items were administered on a 5-point scale (1 = "Strongly disagree" to 5 = "Strongly agree"). Cronbach's alphas were 0.81 (SR) and 0.72 (SE). SR and SE were measured to assess the criterion and incremental validity of MIRES.

### Body appreciation

Body appreciation was measured with the 10-item Body Appreciation Scale-2 (BAS-2) [27]. The scale assesses the tendency of individuals to accept, respect, and have favorable opinions towards their bodies Responses were measured on a 5-point scale (1 = "Never" to 5 = "Always"). Its Cronbach's alpha was 0.96. BAS-2 was measured to assess the criterion and incremental validity of MIRES.

### Self-esteem

To assess self-esteem, we used the Single-Item Self-Esteem scale (SISE) [28], which consists of a single item "I have high self-esteem" administered on a 5-point scale (1 = "Not very true of me" to 5 = "Very true of me"). Using test-retest data over three points in time and following the procedure suggested by Heise [29], developers have obtained a reliability score of 0.75 for SISE. The scale's reliability was not estimated in this study due to the lack of repeated measurements. SISE was measured to assess the criterion and incremental validity of MIRES.

### Life satisfaction

The 5-item Satisfaction With Life Scale (SWLS) [30] was used to measure global cognitive judgments of one's life satisfaction. Items were administered on a 7-point scale (1 = "Strongly disagree" to 7 = "Strongly agree"). Cronbach's alpha was 0.92. SWLS was measured to assess the criterion and incremental validity of MIRES.

### Weight-related measures

Current weight and height were reported in pounds and feet/inches, respectively. Values were transformed to kilograms and meters and were used to calculate Body Mass Index (BMI). Highest and lowest weight during the last four years, excluding periods of pregnancy or sickness, was also reported. Based on subtraction of these values a variable called Maximal Weight Change (MWC) was calculated. Individuals whose MWC was <4kg were classified as with stable weight. Individuals whose MWC was ≥4kg were asked additional questions on their weight trajectory and were categorized into 1. those who gained weight (≥4kg increase in weight without significant fluctuations; fluctuations of ≥4kg were considered significant), 2. those who lost weight (≥4kg decrease in weight without significant fluctuations; fluctuations of ≥4kg were considered significant), or 3. those whose weight cycled (weight had fluctuated

with gains and losses of ≥4kg). Weight cyclers also reported number of intentional weight losses and unintentional weight gains of ≥4kg during the last four years. Responses were used to calculate a measure of Weight Cycling Severity (WCS). These measures were also measured to assess the criterion and incremental validity of MIRES.

## Analysis and results

To confirm the scale's internal structure with CFA and to test several properties of its subscales (i.e., internal consistency, discriminant validity, measurement invariance, construct validity) we administered MIRES to a nearly representative sample (in terms of gender and age) of UK adults (N = 1380) that was recruited via a market research agency (exclusion criteria were pregnancy and lactation, history of eating disorders, diabetes, or bariatric surgery, and current use of appetite-enhancing or -suppressing medication). Data were checked for violations of normality (acceptable skewness values were below 2 in absolute value and acceptable excess kurtosis values below 3 in absolute value) and presence of multivariate outliers (i.e., values outside the boxplots of the Mahalanobis distances for raw scores and residuals). No violations of normality were observed for the variables. After exclusion of multivariate outliers (N = 20) and those who failed an attention check question (N = 386) the sample was skewed towards females and older individuals (Table 1). Given that 195 parameters were to be estimated in the CFA model, the sample size (N = 974) was adequate to get reliable estimates based on the 5:1 partic-ipants-to-parameter ratio [31].

### Internal structure and consistency

The Lavaan package [32] in R (version 3.4.1) [33] was used to conduct CFA with the Maxi-mum Likelihood estimation. Adequacy of fit was determined by four indices (CFI > 0.95, TLI > 0.95, RMSEA < 0.06, SRMR < 0.08) [34]. The structure of MIRES was examined in a sequential process in which individual first-order factor models were tested before subscales were combined into higher-order constructs. The multi-factor model including all MIRES subscales provided a very good fit to the data ($\chi^2$ (1040) = 2567.43, p < 0.001, CFI = 0.97, TLI = 0.97, RMSEA = 0.04, SRMR = 0.04) and all standardized factor loadings were high (above 0.70) and significant (S1 Table). A number of measurement-model modifications were made when testing this model. First, because the items in the sensitivity and self-efficacy sub-scales were asked in triple (across three contexts), method effects were accounted for by allow-ing error terms between identical items to be correlated. Second, because the conceptual distinction between contexts re-appeared in the sensitivity and self-efficacy subscales, we also accounted for context effects by allowing the disturbance terms of the first-order factors refer-ring to the same context to correlate with each other (e.g., SH: Neutral, SS: Neutral, SEH: Neu-tral, SES: Neutral). Composite reliabilities and Average Variance Extracted (AVE) were calculated according to Fornell and Larcker [35]. Reliabilities of the MIRES first- and second-order factors ranged between 0.84 and 0.96, and AVE was as low as 0.64 and as high as 0.88 (Table 2).

### Discriminant validity of constructs

Several alternative models were fitted and compared to show the discriminant validity of the sensitivity and self-efficacy constructs (Table 3). First, to test whether sensitivity and self-effi-cacy are truly distinct from each other we compared two pairs of alternative models: one for hunger and one for satiation. Starting with hunger, in one model the three SH subscales (SH: Neutral, SH: Emotional, SH: External) loaded on a second-order factor SH and the three SEH subscales (SEH: Neutral, SEH: Emotional, SEH: External) loaded on another second-order

**Table 2. Descriptive statistics, composite reliabilities, and AVE for the MIRES first- and second-order factors.**

|  | M | *SD* | Composite reliability | AVE |
|---|---|---|---|---|
| **First-order factors** | | | | |
| IT | 4.52 | 1.68 | 0.94 | 0.80 |
| FL | 4.43 | 1.79 | 0.91 | 0.71 |
| FE | 5.34 | 1.32 | 0.94 | 0.75 |
| SH: Neutral | 5.91 | 1.10 | 0.88 | 0.70 |
| SH: Emotional | 5.38 | 1.48 | 0.88 | 0.71 |
| SH: External | 5.32 | 1.43 | 0.87 | 0.70 |
| SS: Neutral | 5.55 | 1.35 | 0.91 | 0.77 |
| SS: Emotional | 4.83 | 1.73 | 0.89 | 0.73 |
| SS: External | 5.09 | 1.53 | 0.89 | 0.72 |
| SEH: Neutral | 5.49 | 1.34 | 0.90 | 0.75 |
| SEH: Emotional | 4.85 | 1.64 | 0.94 | 0.84 |
| SEH: External | 5.00 | 1.50 | 0.90 | 0.74 |
| SES: Neutral | 5.34 | 1.58 | 0.96 | 0.88 |
| SES: Emotional | 4.69 | 1.87 | 0.91 | 0.76 |
| SES: External | 5.03 | 1.65 | 0.93 | 0.82 |
| **Second-order factors** | | | | |
| SH | 5.54 | 1.14 | 0.84 | 0.64 |
| SS | 5.15 | 1.39 | 0.92 | 0.79 |
| SEH | 5.11 | 1.31 | 0.88 | 0.72 |
| SES | 5.02 | 1.57 | 0.93 | 0.82 |

IT: Internal trust, FL: Food legalizing, FE: Food enjoyment, SH: Sensitivity to physiological signals of hunger, SS: Sensitivity to physiological signals of satiation, SEH: Self-efficacy in using physiological signals of hunger, SES: Self-efficacy in using physiological signals of satiation, AVE: Average Variance Extracted.

factor SEH. In the alternative model, the two second-order factors were collapsed into one factor. The alternative model had significantly lower fit. The same was the case for the distinction between SS and SES.

**Table 3. Change in chi square and fit indices between models testing the discriminant validity of MIRES constructs.**

| Factors[*] | Δχ² (df)[a] | P value | ΔCFI | ΔTLI | ΔRMSEA | ΔSRMR |
|---|---|---|---|---|---|---|
| **Sensitivity vs. Self-efficacy** | | | | | | |
| SH *vs.* SEH | 130.72 (1) | < 0.001 | -0.009 | -0.012 | 0.01 | 0.005 |
| SS *vs.* SES | 116.95 (1) | < 0.001 | -0.006 | -0.008 | 0.011 | 0.005 |
| **Hunger vs. Satiation** | | | | | | |
| SH *vs.* SS | 316.95 (1) | < 0.001 | -0.022 | -0.031 | 0.024 | 0.016 |
| SEH *vs.* SES | 455.77 (1) | < 0.001 | -0.024 | -0.034 | 0.031 | 0.029 |
| **Neutral context vs. Emotional context vs. External context** | | | | | | |
| SH: Neutral *vs.* SH:Emotional *vs.* SH:External | 1341.51 (3) | < 0.001 | -0.235 | -0.47 | 0.276 | 0.086 |
| SS: Neutral *vs.* SS:Emotional *vs.* SS:External | 1005.99 (3) | < 0.001 | -0.139 | -0.278 | 0.211 | 0.048 |
| SEH: Neutral *vs.* SEH:Emotional *vs.* SEH:External | 1300.46 (3) | < 0.001 | -0.188 | -0.377 | 0.239 | 0.065 |
| SES: Neutral *vs.* SES:Emotional *vs.* SES:External | 1633.31 (3) | < 0.001 | -0.158 | -0.315 | 0.267 | 0.051 |

SH: Sensitivity to physiological signals of hunger, SS: Sensitivity to physiological signals of satiation, SEH: Self-efficacy in using physiological signals of hunger, SES: Self-efficacy in using physiological signals of satiation.

[*] In the initial model, factors were distinct. In the alternative model, factors were collapsed into a single factor.

[a] Alternative model–Initial model.

In a similar way, we tested the discriminant validity of hunger and satiation constructs by comparing two pairs of alternative models: one for sensitivity and one for self-efficacy. The alternative model, in which SH and SS were collapsed into one factor, was significantly worse compared to the model where the two factors were distinct. The same was the case for SEH and SES.

Finally, the conceptual distinction between different contexts of sensitivity and self-efficacy was tested. For each second-order construct (SH, SS, SEH, and SES), we compared the fit of a three-factor model in which each item loaded to its respective context versus an alternative model in which the three factors were collapsed into one factor. In all cases, the fit of the alternative model was significantly worse.

## Measurement invariance

Measurement invariance was examined for the items that were asked in triple (across contexts) to test the assumption that each item should have a consistent performance irrespectively of the context in which it is asked. To do this, we constrained the loadings of these items to be equal across the three contexts. The decrease in fit in the constrained model was significant ($\Delta\chi^2$ (24) = 102.502, p < 0.001), however, the changes in fit indices were within the acceptable criteria ($\Delta$CFI = -0.002, $\Delta$TLI = -0.001, $\Delta$RMSEA = 0, $\Delta$SRMR = 0.001) according to Chen's [36] recommendations for factor loading invariance ($\Delta$CFI $\leq$ 0.010, $\Delta$RMSEA $\leq$ 0.015, and $\Delta$SRMR $\leq$ 0.030).

## Construct validity

Since the IRE style is by nature a non-diet eating style, we used independent samples t-tests to compare scores on the MIRES subscales between individuals who said they were currently dieting for weight loss purposes ($n_1$ = 131) and those who said they were not ($n_2$ = 843), as a means of testing the scale for construct validity in a broad sense. Non-dieters scored significantly higher than dieters in all but one MIRES subscales, in line with our expectations (S2 Table). For FE, the mean difference between groups did not reach significance.

## Temporal stability

A sub-sample of 679 participants from the UK sample filled in the MIRES for a second time after two weeks. Response rate was 43.2%, but the entire survey was completed by 261 participants. Those who failed the attention check (N = 46) and two multivariate outliers were excluded, leaving a sample of 213 responses for analysis (Table 1). The sample size was adequate to get reliable estimates in models testing the stability of first-order factors, while in models testing the stability of second-order factors the sample was slightly small (4:1 participant-to-parameter-ratio).

No violations of normality were observed for the variables. We used an elaborated procedure of temporal stability assessment as suggested by Steenkamp and van Trijp [37]. Pearson's correlation coefficients, intra-class coefficients with confidence intervals, and means for the summed scores of factors were also calculated. Stability coefficients of the MIRES first- and second-order factors ranged between .63 and .90 (Table 4). Imposition of constraints on factor loadings did not result in significant decreases in model fit, thus, the meaning of all subscales was stable. Some subscales were further found to be stable in terms of item reliabilities (SS: Neutral and EH: External) and construct reliability (FL, SH: External, SS: Emotional, SS: External, EH: Emotional, and ES: Neutral). Finally, SH: Neutral, SEH: Neutral, and SEH manifested perfect stability as their stability coefficient was not significantly different from unity. Paired samples t-tests indicated that most factor means were stable over time; however, the means of IT, FL, SH: Emotional, and SS: External changed significantly.

**Table 4. Stability coefficients, Pearson correlation coefficients, intra-class correlation coefficients, and mean scores for the MIRES first- and second- order factors.**

|  | Stability coefficient | Pearson's r | ICC (CI)** | Mean 1 | Mean 2 | P value |
|---|---|---|---|---|---|---|
| **First-order factors** |  |  |  |  |  |  |
| IT | 0.74 | 0.69* | 0.80 (0.73–0.86) | 18.64 | 20.22 | < 0.001 |
| FL | 0.79 | 0.74* | 0.85 (0.80–0.89) | 18.39 | 19.37 | 0.005 |
| FE | 0.67 | 0.65* | 0.79 (0.72–0.84) | 27.00 | 27.42 | 0.292 |
| SH: Neutral | 0.66 | 0.57* | 0.73 (0.64–0.79) | 17.79 | 17.90 | 0.624 |
| SS: Neutral | 0.74 | 0.69* | 0.81 (0.75–0.86) | 17.20 | 17.08 | 0.556 |
| SH: Emotional | 0.69 | 0.64* | 0.77 (0.70–0.83) | 16.63 | 16.08 | 0.037 |
| SS: Emotional | 0.83 | 0.77* | 0.87 (0.83–0.90) | 15.16 | 15.07 | 0.702 |
| SH: External | 0.70 | 0.62* | 0.77 (0.69–0.82) | 16.21 | 15.82 | 0.133 |
| SS: External | 0.76 | 0.70* | 0.82 (0.76–0.86) | 16.08 | 15.56 | 0.033 |
| SEH: Neutral | 0.63 | 0.59* | 0.74 (0.66–0.80) | 16.84 | 16.92 | 0.754 |
| SES: Neutral | 0.76 | 0.71* | 0.83 (0.78–0.87) | 16.84 | 16.87 | 0.903 |
| SEH: Emotional | 0.65 | 0.61* | 0.76 (0.68–0.82) | 15.30 | 14.88 | 0.161 |
| SES: Emotional | 0.74 | 0.71* | 0.83 (0.78–0.87) | 15.24 | 14.79 | 0.125 |
| SEH: External | 0.71 | 0.65* | 0.78 (0.71–0.83) | 15.47 | 15.09 | 0.148 |
| SES: External | 0.72 | 0.68* | 0.81 (0.75–0.85) | 15.96 | 15.62 | 0.204 |
| **Second-order factors** |  |  |  |  |  |  |
| SH | 0.90 | 0.75* | 0.85 (0.81–0.89) | 50.63 | 49.79 | 0.102 |
| SS | 0.90 | 0.83* | 0.90 (0.87–0.93) | 48.45 | 47.70 | 0.145 |
| SEH | 0.83 | 0.71* | 0.83 (0.78–0.87) | 47.62 | 46.89 | 0.237 |
| SES | 0.85 | 0.78* | 0.88 (0.84–0.91) | 48.05 | 47.27 | 0.218 |

IT: Internal trust, FL: Food legalizing, FE: Food enjoyment, SH: Sensitivity to physiological signals of hunger, SS: Sensitivity to physiological signals of satiation, SEH: Self-efficacy in using physiological signals of hunger, SES: Self-efficacy in using physiological signals of satiation.

* p < 0.001.

** Intra-class correlation coefficients using an absolute agreement definition.

## Length optimization

In order to further optimize the scale's length and to have the same number of items per subscale (i.e., three), we decided to drop seven items; four items from the IT subscale, one item from the FL subscale, and two items from the FE subscale. The decision on which items to drop was based on the meaning of items to retain the scale's content validity [38]; items whose meaning was very similar to other items in their respective subscales were dropped. The three subscales manifested similar properties after the exclusion of items (IT: Stability coefficient = 0.70, r = 0.65, ICC = 0.78 (0.70–0.84), Mean 1 = 14.00, Mean 2 = 15.16, p < 0.001; FL: Stability coefficient = 0.82, r = 0.74, ICC = 0.85 (0.80–0.88), Mean 1 = 13.72, Mean 2 = 14.46, p = 0.005; FE: Stability coefficient = 0.66, r = 0.61, ICC = 0.76 (0.69–0.82), Mean 1 = 16.01, Mean 2 = 16.33, p = 0.204). The final scale consisted of 45 items.

## Confirmation of the internal structure of MIRES as a multidimensional, formative model

The 45-item MIRES was further administered to a representative sample of 1251 adults from the US [39] (Table 1; see also S3 Table for some additional characteristics) (recruited via a market research agency) in order to confirm the internal structure of MIRES as a multidimensional formative model and to test the scale's convergent, criterion, and incremental validity.

Exclusion criteria were pregnancy and lactation, because these conditions relate to temporal irregularities in the eating patterns of women. Fifty-one multivariate outliers were excluded leaving 1200 responses for analysis. Based on the recommended 5:1 participants-to-parameter ratio, a sample of 1200 participants would be adequate to give reliable estimates for a model with maximum 240 parameters. All models that we tested had less than 240 parameters to be estimated, thus the sample size was adequate for our analyses. No significant violations of normality were observed for most variables. BMI and MWC had kurtosis values above 3 and the latter also had a skewness value above 2. However, according to Kline's [40] more relaxed criteria for skewness and kurtosis (<3 and <10, respectively) none of these variables were considered problematic, thus no transformations were conducted.

The MIRES model was subjected to CFA (S2 Fig) with the following additional specifications. The three first-order factors—IT, FL, FE—and the four second-order factors—SH, SS, SEH, SES—loaded to the higher-order IRE style construct as formative indicators (arrows pointing to the higher-order construct). Covariances between all first- and second-order factors with the higher-order formative factor were fixed to zero, as otherwise Lavaan estimates both these covariances and the formative regression coefficients, which seem to be confounded leading to identification problems. To warrant identification, the six RI also loaded to the IRE style construct as reflective indicators (arrows pointing to the six RI).

The model had an excellent fit to the data ($\chi^2$ (1130) = 2804.10, p < 0.001, CFI = 0.97, TLI = 0.97, RMSEA = 0.04, SRMR = 0.03). All observed variables served as reliable and significant indicators of their corresponding constructs and all first-order factors loaded highly and significantly to their respective second-order factors (S2 Fig), as was the case in the UK sample. Regression coefficients of the seven formative indicators of the IRE style were not interpreted because their values were influenced by the presence of multi-collinearity among the seven subscales of MIRES (Variance Inflation Factors 1.52–7.85, cut-off <3.3), which are moderately to strongly correlated with each other (S4 Table). High and significant loadings were obtained for the six RI (0.66–0.86) and a large amount of variance in these items was accounted for by the IRE style factor (AVE = 0.82).

## Convergent validity

Bivariate correlations of the MIRES total score, RI, and MIRES subscales with the IES-2 and ecSI-2 total scores were substantial and significant (0.32–0.70) (S5 Table). High correlations were particularly observed between certain MIRES subscales and conceptually related constructs of IES-2 and ecSI-2. For example, FL and FE correlated most strongly with the EatAtt (0.56) and ContSkills (0.46) subscales of ecSI-2, respectively. Similarly, SEH and SES correlated most strongly with the RHSC subscale of IES-2 (0.66 and 0.68, respectively).

## Criterion validity

The criterion validity of MIRES, IES-2, and ecSI-2 was examined with Structural Equation Modelling (SEM) (for outcomes measured with multiple items) and with linear regression (for the single-item outcomes SISE, BMI, MWC, and WCS). Analyses with MIRES were conducted at the level of a total score (summed score of all items), at the level of the seven MIRES subscales as separate latent constructs (IT, FL, FE, SH, SS, SEH, SES), and at the level of the RI as an independent scale. Analyses for IES-2 and ecSI-2 were conducted only at the level of total scores.

MIRES, as well as its individual subscales, displayed negative associations with binge eating, restrictive eating, BMI, maximal weight change, and weight cycling severity, and positive associations with all adaptive outcomes assessed in this study (Table 5). In general, MIRES, IES-2,

**Table 5. Bivariate correlations among all constructs measured in the US sample.**

| | 1 | 2 | 3 | 4 | 5 | 6 | 7 | 8 | 9 | 10 | 11 | 12 | 13 | 14 |
|---|---|---|---|---|---|---|---|---|---|---|---|---|---|---|
| **1. MIRES** | - | | | | | | | | | | | | | |
| **2. IES-2** | 0.69** | - | | | | | | | | | | | | |
| **3. ecSI-2** | 0.67** | 0.60** | - | | | | | | | | | | | |
| **4. BES** | -0.38** | -0.46** | -0.16** | - | | | | | | | | | | |
| **5. RES** | -0.15** | -0.12** | -0.12** | 0.47** | - | | | | | | | | | |
| **6. PCS** | 0.38** | 0.35** | 0.44** | -0.02 | 0.14** | - | | | | | | | | |
| **7. SR** | 0.23** | 0.19** | 0.15** | 0.002 | 0.34** | 0.27** | - | | | | | | | |
| **8. SE** | 0.23** | 0.21** | 0.17** | -0.14** | 0.16** | 0.21** | 0.39** | - | | | | | | |
| **9. BAS-2** | 0.49** | 0.53** | 0.59** | -0.26** | -0.02 | 0.52** | 0.26** | 0.24** | - | | | | | |
| **10. SWLS** | 0.29** | 0.28** | 0.40** | -0.04 | 0.04 | 0.44** | 0.24** | 0.16** | 0.62** | - | | | | |
| **11. SISE** | 0.34** | 0.35** | 0.40** | -0.15** | 0.003 | 0.40** | 0.19** | 0.12** | 0.71** | 0.60** | - | | | |
| **12. BMI** | -0.15** | -0.21** | -0.12** | 0.20** | 0.05 | -0.09** | -0.12** | -0.07** | -0.21** | -0.10** | -0.12** | - | | |
| **13. MWC** | -0.16** | -0.21** | -0.16** | 0.13** | 0.16** | -0.08** | 0.002 | 0.005 | -0.19** | -0.13** | -0.12** | 0.43** | - | |
| **14. WCS** | -0.22** | -0.27** | -0.09 | 0.34** | 0.27** | 0.11* | 0.08 | -0.005 | -0.06 | 0.01 | -0.001 | 0.24** | 0.29** | - |

MIRES: Multidimensional Internally Regulated Eating Scale, IES-2: Intuitive Eating Scale-2, ecSI-2: Eating Competence Satter Inventory 2.0, BES: Binge Eating Scale, RES: Restrictive Eating Scale, PCS: Proactive Coping Scale, SR: Satiety Responsiveness, SE: Slowness in Eating, BAS-2: Body Appreciation Scale-2, SWLS: Satisfaction With Life Scale, SISE: Single Item Self-Esteem Scale, BMI: Body Mass Index, MWC: Maximal Weight Change, WCS: Weight Cycling Severity.

* Correlation is significant at the 0.05 level.

** Correlation is significant at the 0.01 level.

and ecSI-2 displayed comparable predictive abilities (S6 Table) and all were better at predicting behavioral and psychological outcomes, compared to physical outcomes. MIRES accounted for a slightly larger amount of variance in RES, SR, and SE compared to the other scales, IES-2 was better at predicting BES, BMI, MWC, and WCS, and finally ecSI-2 was better at predicting PCS, BAS-2, SWLS, and SISE. The RI manifested comparable criterion validity to MIRES. Finally, certain MIRES subscales (FL, SH, SS, SES) achieved higher predictive power compared to the MIRES summed score for certain outcomes (e.g., RES, BES, SR, SE, BMI).

## Incremental validity

The incremental validity of MIRES in relation to IES-2 and ecSI-2 was examined with SEM (for multi-item outcomes) and hierarchical regression analysis (for single-item outcomes). Specifically, we examined whether MIRES accounted for variance in each outcome measure above and beyond the variance accounted for by IES-2 and ecSI-2, respectively. At Step 1, IES-2 was entered as a single predictor of each respective outcome and at Step 2, MIRES was added as a second predictor (in SEM analyses, MIRES was also entered as a predictor in the model at Step 1, but its regression coefficient was fixed at zero). The same procedure was followed with ecSI-2. Changes in beta coefficients were not interpreted because multi-collinearity between these conceptually similar measures was expected to interfere with these estimates. For most outcomes, a significant increase in $R^2$ was observed when MIRES was added in the model (Table 6). Specifically, MIRES accounted for 0.7%-16% additional variance in outcome measures above and beyond IES-2 and ecSI-2. MIRES did not account for a significant increase in explained variance of physical outcomes (BMI [$\Delta R^2 = 0$], MWC [[$\Delta R^2 = 0$], and WCS [[$\Delta R^2 = 0.002$]) above and beyond IES-2, neither for satisfaction with life ($\Delta R^2 = 0$) and self-esteem ($\Delta R^2 = 0.005$) above and beyond the variance explained for by ecSI-2.

**Table 6. Incremental variance in outcome measures accounted for by MIRES.**

| | MIRES vs. IES-2 | | | | MIRES vs. ecSI-2 | | | |
|---|---|---|---|---|---|---|---|---|
| | $R^2$ (IES-2) | $R^2$ (IES-2 + MIRES) | $\Delta R^2$ | P value | $R^2$ (ecSI-2) | $R^2$ (ecSI-2 + MIRES) | $\Delta R^2$ | P value |
| BES[a] | 0.249 | 0.259 | 0.010 | <0.001 | 0.032 | 0.192 | 0.160 | <0.001 |
| RES[a] | 0.021 | 0.030 | 0.009 | 0.001 | 0.018 | 0.030 | 0.012 | <0.001 |
| PCS[a] | 0.150 | 0.191 | 0.041 | <0.001 | 0.227 | 0.244 | 0.017 | <0.001 |
| SR[a] | 0.047 | 0.069 | 0.022 | <0.001 | 0.028 | 0.066 | 0.038 | <0.001 |
| SE[a] | 0.048 | 0.072 | 0.024 | <0.001 | 0.052 | 0.074 | 0.022 | <0.001 |
| BAS-2[a] | 0.284 | 0.316 | 0.032 | <0.001 | 0.348 | 0.367 | 0.019 | <0.001 |
| SWLS[a] | 0.085 | 0.104 | 0.019 | <0.001 | 0.178 | 0.178 | 0.000 | 0.370 |
| SISE[b] | 0.119 | 0.138 | 0.019 | <0.001 | 0.156 | 0.161 | 0.005 | 0.085 |
| BMI[b] | 0.045 | 0.045 | 0.000 | 0.802 | 0.013 | 0.043 | 0.03 | <0.001 |
| MWC[b] | 0.043 | 0.043 | 0.000 | 0.485 | 0.035 | 0.042 | 0.007 | 0.047 |
| WCS[b*] | 0.071 | 0.073 | 0.002 | 0.298 | 0.007 | 0.055 | 0.048 | <0.001 |

MIRES: Multidimensional Internally Regulated Eating Scale, IES-2: Intuitive Eating Scale-2, ecSI-2: Eating Competence Satter Inventory 2, BES: Binge Eating Scale, RES: Restrictive Eating Scale, PCS: Proactive Coping Scale, SR: Satiety Responsiveness, SE: Slowness in Eating, BAS-2: Body Appreciation Scale-2, SWLS: Satisfaction With Life Scale, SISE: Single Item Self-Esteem Scale, BMI: Body Mass Index, MWC: Maximal Weight Change, WCS: Weight Cycling Severity.

[a] Values obtained with SEM.

[b] Values obtained with hierarchical regression analysis.

\* N = 504.

## Testing the properties of the simplified 21-item version of MIRES

Since the 45-item MIRES manifested good psychometric properties, we wanted to examine whether the inclusion of the three contexts (neutral, emotional, external) in the sensitivity and self-efficacy subscales offers predictive advantages compared to just the neutral context. In this way we could ascertain whether a simplified version of the scale (21 items) could still be applicable. To test this empirically we performed SEM and regression analysis (depending on the outcome variable) using either the full subscales (SH, SS, SEH, and SES) including all three contexts each or the neutral counterpart of each subscale to predict each outcome measured in the US sample. The full subscales accounted for 0–8% additional variance, depending on the outcome, compared to their neutral counterparts (S7 Table). In addition, the fit of the 21-item MIRES model was still excellent ($\chi^2$ (296) = 1258.161, p < 0.001, CFI = 0.97, TLI = 0.96, RMSEA = 0.05, SRMR = 0.04) (Fig 2), correlations among the MIRES subscales and with IES-2 and ecSI-2 reduced only slightly (S8 and S9 Tables), and the incremental validity of MIRES was still upheld (S10 Table). Thus, despite the fact that the 45-item full version offers some predictive advantages, the simplified version with only 21 items generally upholds the psychometric properties of the full scale.

## Discussion

Internally regulated eating is an adaptive way of eating that leads to positive physical, psychological, behavioral, and dietary outcomes as shown by the current and previous research [1–6]. While several attempts have been made to conceptualize and quantify this eating style, none seems to capture the full complexity of this construct. In this paper, we describe the rigorous development and validation of the MIRES, an instrument to assess the individual-difference characteristics that are necessary and jointly sufficient conditions for the manifestation of the IRE style.

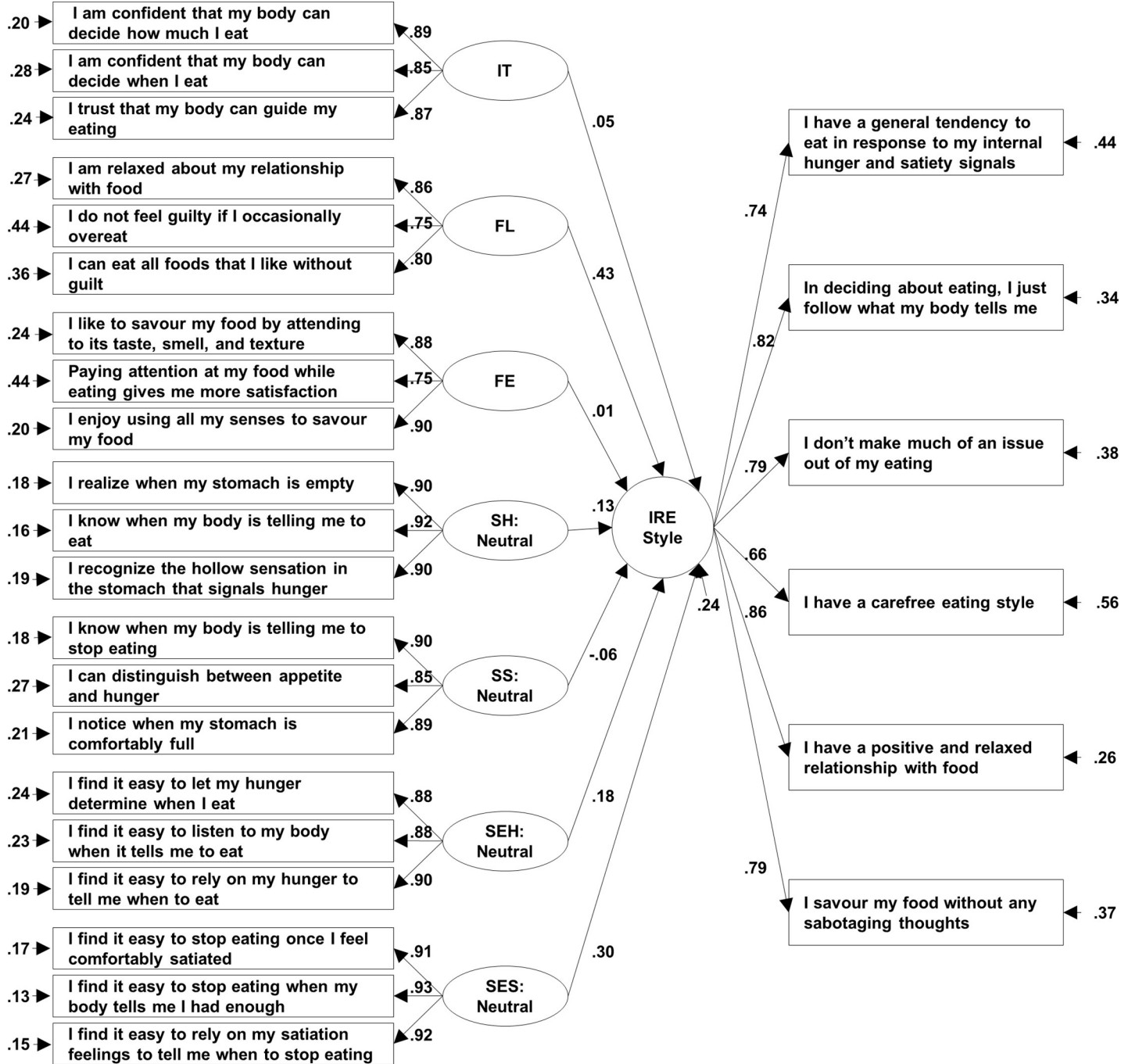

**Fig 2. The multi-dimensional model of internally regulated eating style (simplified version).** All loadings were significant at the 0.01 level. Covariances and disturbance terms of first-order factors are not depicted in the figure for easier readability.

Using a bottom-up approach, we showed that all first- and second-order factors of MIRES are measured reliably and a significant amount of variance in the items is accounted for by the corresponding latent factors. All first-order models and the multi-factor model that we tested had very good fit to the data. We confirmed that sensitivity to hunger, sensitivity to satiation, self-efficacy with hunger, and self-efficacy with satiation are distinct constructs, and that the

three contexts within each of these subscales are also distinct from each other. Results supported the metric measurement invariance of the items asked across contexts and initial evidence on the construct validity of MIRES was obtained, as non-dieters scored higher in all but one MIRES subscales compared to dieters. Scores on FE did not differ significantly between groups, suggesting that this is perhaps the least determinative characteristic among the ones that form the IRE style. We further showed that all MIRES subscales are stable over a period of two weeks in terms of factor loadings, while even higher levels of stability (in terms of item reliabilities, construct reliabilities, or correlation of the same factor over time) were evidenced for certain subscales. Pearson's correlations underestimated the true stability of these constructs, while intra-class correlation coefficients overestimated it. Factor means remained stable for most factors except for IT, FL, SH: Emotional, and SS: External. As regards the latter two factors, however, the means of their respective second-order factors (SH and SS) were stable. The change in means in IT and FL, suggests that these subscales show variation over time across the whole sample, which could be systematic (i.e., these subscales measure less stable characteristics) or random (i.e., due to chance). Further studies are required to confirm which of the two plausible explanations is true. Evidence on the multidimensional nature of the MIRES model was also obtained in this study. The convergent validity of MIRES was supported by the moderate to strong correlations with measures of intuitive eating and eating competence. Measures of IRE were generally better at predicting behavioral and psychological outcomes compared to physical outcomes, which is in line with existing evidence [1–3, 6]. MIRES associated negatively with binge eating, restrictive eating, BMI, maximal weight change, and weight cycling severity, and positively with all adaptive outcomes assessed in this study. This confirms the adaptive nature of the constructs it assesses. The six RI had comparable predictive power to the 45-item MIRES. Furthermore, certain MIRES subscales (FL, SH, SS, and SES) accounted for a larger amount of variance in certain outcomes compared to the MIRES summed score. This further justifies their applicability as independent measures. The incremental validity of MIRES, above and beyond IES-2 and ecSI-2, was supported for most outcome variables measured in this study. Finally, we showed that the simplified 21-item version of MIRES upholds the psychometric properties of the full 45-item scale.

MIRES can be used by researchers and practitioners for a complete assessment of the IRE style as well as of its distinct components. MIRES can be used as an independent variable, moderator, or mediator in future scientific research investigating the role of IRE style in various processes in the eating domain. It can also be used as an outcome variable when assessing the impact of interventions aimed to strengthen IRE. Finally, MIRES can be used as a screening instrument by health practitioners who try to promote IRE among their clients or patients.

While MIRES manifested good psychometric properties, there are limitations that should be addressed. First, we should note that all data presented in this paper are solely based on self-reports. Although self-reports are practical tools for the assessment of personality constructs, they are subject to several types of response bias such as socially desirable responding, acquiescent responding, or extreme responding [41]. Individual responses may also be limited by the lack of sufficient self-awareness or by self-deception effects. Second, identification restrictions are inherent to formative models [42], as is the one presented in this paper. Thus, researchers who are interested in conducting CFA or SEM using the complete formative MIRES model should also measure the six RI that we specifically developed to facilitate model identification. Third, the preliminary work was conducted with college students (18–35 years old) while in later steps we used community samples (18–65 years old); thus, it could be argued that it is not safe to assume the invariance of the model's internal structure across the scale development and validation process. To test the model for measurement invariance across age groups, subgroups should have at least 980 participants each to allow for reliable estimates to emerge

based on the 5:1 participant to parameter ratio. The sample sizes in our study did not allow us to conduct this analysis in the typical stepwise process [43]; however, when we fitted the model in subgroups with all but seven parameters fixed to the values obtained from the full sample (only regression coefficients of the seven formative indicators were left free to be estimated) the model fit was still acceptable (18–34 years: $\chi^2$ (1319) = 2467.93, p < 0.001, CFI = 0.95, TLI = 0.95, RMSEA = 0.05, SRMR = 0.03; 35–65 years: $\chi^2$ (1319) = 2969.25, p < 0.001, CFI = 0.96, TLI = 0.96, RMSEA = 0.04, SRMR = 0.05) providing, thus, preliminary evidence for the invariance of the model across age groups. Finally, we acknowledge that administration of the full version of MIRES may be more complex than other self-reports because twelve of its items are repeated across three different contexts. Thus, we advise potential users to use the simplified version of the scale that consists of only 21 items.

Next to these limitations, the strengths of this newly developed measure should also be considered. In contrast to what most scale developers do, in this research we were particularly interested in the precise specification of the measurement model. Those who aim to assess the IRE style need to measure the complete set of seven MIRES subscales and calculate a total score, while those who want to focus on a particular characteristic of the IRE style can choose to measure a subscale in isolation and calculate the summed score of items of that particular subscale. The bottom-up approach that we took for the scale's development and validation (assessing the properties of lower-order factors before moving to higher levels) can give researchers and practitioners confidence on the reliability and validity of the scale's sub-parts. It should be noted here that using only a subset of subscales would allow conclusions to be drawn only on those particular constructs that are measured and not on the IRE style construct. We further observed strong convergence and comparable criterion validity between MIRES and the six RI. Given that RI is a reliable scale in itself, it could be used as the snap version of MIRES. This adds even more flexibility in the use of the new instrument. Finally, the multidimensional nature of MIRES enables the distinction of several closely related but conceptually distinct features of the IRE style. For example, the distinction between sensitivity to and self-efficacy in using physiological signals of hunger and satiation has been examined very deficiently in existing literature (see e.g., [44]). Therefore, MIRES can be used for a more differentiated assessment of the essentials of the IRE style.

Although we followed a rigorous process for the scale's development and validation, replication of the current findings in other populations or population segments is needed. For example, the measurement invariance of the model could be tested across sexes, age groups, and other potentially interesting population groups such individuals with overweight or obesity. Once measurement invariance of the model is evidenced, norm scores can be developed for the various subgroups. Moreover, it would be interesting to administer the simplified version of the scale without any introductory text in the sensitivity and self-efficacy subscales in order to ascertain whether this influences how individuals interpret the items. Additional studies could also be conducted to assess the temporal stability of the RI scale and to ascertain whether the change in means over time in two MIRES subscales (IT and FL) that we observed was systematic or random. Future research could also test the face validity of the final MIRES because relevance of items with the construct definitions was assessed only at the very beginning of the scale development process. This would ensure that the retained items still do a good job in reflecting the meaning of the constructs they are purported to measure. Given that a theory-based approach was used in this research, we expect that MIRES will uphold its face validity. Finally, behavioral experiments could provide convincing and invaluable evidence for the construct and predictive validity of MIRES.

## Supporting information

**S1 Fig. Conceptual model of internally regulated eating style.** The direction of arrows indicates whether a construct is formative—arrows point to the construct—or reflective—arrows point to the dimension.
(TIF)

**S2 Fig. The multi-dimensional model of internally regulated eating style (full version).** All loadings were significant at the 0.01 level. Context effects, method effects, covariances between first- and second-order factors, and disturbance terms of first- and second-order factors are not depicted in the figure for easier readability.
(TIF)

**S1 Table. Factor loadings for the MIRES first- and second-order factors.**< /SI_Caption>
(DOCX)

**S2 Table. Mean scores on MIRES first- and second-order factors for dieters and non-dieters.**
(DOCX)

**S3 Table. Additional sample characteristics of the US sample.**
(DOCX)

**S4 Table. Bivariate correlations of summed scores of MIRES, RI, and MIRES subscales.**
(DOCX)

**S5 Table. Bivariate correlations of summed scores of MIRES, RI, and MIRES subscales with IES-2 and ecSI-2.**
(DOCX)

**S6 Table. Standardized regression coefficients (and $R^2$) for the criterion validity of MIRES, IES-2, and ecSI-2.**
(DOCX)

**S7 Table. Standardized regression coefficients (and $R^2$) for the criterion validity of SH, SS, SEH, SES (full subscales including neutral, emotional, external contexts) vs. the neutral counterpart of each subscale.**
(DOCX)

**S8 Table. Bivariate correlations of summed scores of MIRES (21 items), RI, and MIRES subscales.**
(DOCX)

**S9 Table. Bivariate correlations of summed scores of MIRES (21 items), RI, and MIRES subscales with IES-2 and ecSI-2.**
(DOCX)

**S10 Table. Incremental variance in outcome measures accounted for by MIRES (21 items).**
(DOCX)

**S1 Appendix. The Multidimensional Internally Regulated Eating Scale (MIRES).**
(DOCX)

**S2 Appendix. Modifications of the MIRES item pool during the scale development and validation process.**
(DOCX)

## Acknowledgments

The authors would like to thank Dr. I. van der Lans for his valuable comments on the statistical analyses and the manuscript, as well as all individuals who participated in the studies presented in this paper.

## Author Contributions

**Conceptualization:** Aikaterini Palascha, Ellen van Kleef, Emely de Vet, Hans C. M. van Trijp.

**Data curation:** Aikaterini Palascha.

**Formal analysis:** Aikaterini Palascha.

**Methodology:** Aikaterini Palascha.

**Supervision:** Ellen van Kleef, Emely de Vet, Hans C. M. van Trijp.

**Visualization:** Aikaterini Palascha.

**Writing – original draft:** Aikaterini Palascha.

**Writing – review & editing:** Aikaterini Palascha, Ellen van Kleef, Emely de Vet, Hans C. M. van Trijp.

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
