## [Decision Letter · Decision Letter 0]

13 Dec 2019

PONE-D-19-26656

Development and validation of the Multidimensional Internally Regulated Eating Scale (MIRES)

PLOS ONE

Dear Mrs Palascha,

Thank you for submitting your manuscript to PLOS ONE. After careful consideration, we feel that it has merit but does not fully meet PLOS ONE’s publication criteria as it currently stands. Therefore, we invite you to submit a revised version of the manuscript that addresses the points raised during the review process.

We would appreciate receiving your revised manuscript by Jan 27 2020 11:59PM. To enhance the reproducibility of your results, we recommend that if applicable you deposit your laboratory protocols in protocols.io, where a protocol can be assigned its own identifier (DOI) such that it can be cited independently in the future. For instructions see: http://journals.plos.org/plosone/s/submission-guidelines#loc-laboratory-protocols

We look forward to receiving your revised manuscript.

Kind regards,

Gian Mauro Manzoni, Ph.D., Psy.D.

Academic Editor

PLOS ONE

Journal Requirements:

3. Your ethics statement must appear in the Methods section of your manuscript. If your ethics statement is written in any section besides the Methods, please move it to the Methods section and delete it from any other section. Please also ensure that your ethics statement is included in your manuscript, as the ethics section of your online submission will not be published alongside your manuscript.

4.  Thank you for including your ethics statement:  "The studies were conducted according to the guidelines laid down in the Declaration of Helsinki and complied with the code of conduct of Wageningen University. Written consent was obtained for all survey participants.". 

For studies reporting research involving human participants, PLOS ONE requires authors to confirm that this specific study was reviewed and approved by an institutional review board (ethics committee) before the study began. Please provide the specific name of the ethics committee/IRB that approved your study, or explain why you did not seek approval in this case.

Reviewers' comments:

Reviewer's Responses to Questions

**Comments to the Author**

1. Is the manuscript technically sound, and do the data support the conclusions?

Reviewer #1: Yes

Reviewer #2: No

2. Has the statistical analysis been performed appropriately and rigorously? 

Reviewer #1: Yes

Reviewer #2: No

3. Have the authors made all data underlying the findings in their manuscript fully available?

Reviewer #1: Yes

Reviewer #2: Yes

4. Is the manuscript presented in an intelligible fashion and written in standard English?

Reviewer #1: Yes

Reviewer #2: No

5. Review Comments to the Author

Reviewer #1: The authors present the results of an effort to develop a new measure of internally regulated eating. They make a reasonable case for the need for such a measure given extant measures of associated constructs. They mine the literature for dimensions that conceptually define internally regulated eating and develop a self-report measure that maps closely onto a domain-space defined by those dimensions. Their work is among the best I have read in terms of scale development and initial validation. Item generation and selection are thoughtful, and the psychometric analyses are thorough and informative. All in all, it is, in my view, a strong manuscript.

Here are three minor considerations:

1. It is not clear to my what is to be gained from the series of one-factor models evaluated as described on pp. 12-13. The problem with these analyses is that do not examine the critical question of whether items written for a particular dimension uniquely indicate the intended dimension. It leads to reporting loadings in Table 1 that appear to be from a single model but in fact are from many small models. All I needed to see was the test of the full model, reported around the middle of p. 13, and my preference would be to see the loadings from that model in a supplemental table.

2. I understand the context distinction, but building it into the measures significantly increases the length and complexity of the measure if used as the authors intend. The question is whether information about internally regulated eating from specific contexts offers predictive advantages in hypothesis tests. If, as an example, one looks at the dieters-nondieters comparisons in Table S1, Cohen's ds are virtually identical for SS and SES and not substantially different for SH and SEH. In terms of the latter two, it is notable that the inconsistent d is for the emotional context, which seems to be operating somewhat differently than the other two here and in other results. Similarly, the stability coefficients in Table 4 are very similar within dimension across context (e.g., .57, .64 and .62 for SH). Scores are collapsed across context within dimension in Tables S4 and S5, making it impossible to determine whether predictive or incremental validity varies as a function of context. In short, I think the authors need to make a stronger case for building context into the measure. My recommendation would be to, if at all possible, drop the distinction.

3. One issue for measures like this is how they should be scored. For the validity analyses, the authors consider both a total score and a score by dimension (ignoring context). My inference is that they would favor either scoring depending on research question. It is worth noting that researchers sometimes use only a subset of subscales from a measure like this. When the first-order factors are reflective indicators of the general factor, a total score can be meaningfully interpreted. In the case of formative indicators, as the authors have cast the first-order factors, that is not the case. The general factor is not fully defined unless all of the formative indicators are included in a composite.

A very minor matter: On p. 33, just before the general discussion, the authors suggest differences that imply comparisons of correlation coefficients, which they did not do. "Outperformed" implies a significantly larger coefficient, which requires a test comparing them. One somewhat related and even more minor matter: Please include the actual coefficients for all pairwise rs in Table 5. The use of "NS" in a table like this is a meta-analyst's nightmare and, generally speaking, unnecessarily hides information that need not be hidden.

These matters are very minor and do not detract from my overall positive view of the manuscript. I congratulate the authors on a fine effort.

Reviewer #2: The authors report on the development of a new measure (MIRES). My first impression is that this is an extremely long paper, for its content. As such, it loses focus and clarity about its aims. By the end, I was not clear what psychological or clinical issue the authors were addressing, or what the implications of the work would be.

More importantly, the presentation failed to meet a number of basic psychometric requirements in developing such a measure.

MAJOR ISSUES

There is a core problem with the reporting of the initial samples used in studies 3-7. The ages of the first two samples are substantially different, making the findings hard to compare. Without age-based norms for the original measure, comparability of scores or of factor structure is not safe to assume.

Studies 1-4 are simply not reported in anything like adequate detail. Where are the initial factor analyses that I presume must have been conducted? What factors emerged, using what methods? Was there replication? Internal consistency? Test-retest reliability? Most importantly, how do the authors justify a sample in studies 3 and 4 that are simply too small to allow for reliable factor derivation (80 items requires a sample of 800 participants PER STUDY to ensure reliable outcomes)? In short, the derivation of a pool of 49 items is just not safe. Nor do we have any reason to assume that the factors that are addressed in studies 5-7 are in any way related to what was found in Studies 1-4.

For Study 5, the construct validity element is not clear. What were the t-values per measure, and what was the P value adopted? Why not use binomial regression to determine the key MIRES variables? And why use one simple measure like this (dieting/not dieting), when understanding such a complex construct? An appropriate index would have been to determine whether the MIRES is a better predictor than other measures of the same constructs, to show that the MIRES is a more clinically useful measure.

If Study 6 really did use the same analyses as Study 5, where is the report of the CFA? If one was conducted, then the sample in Study 6 appears to be underpowered, even by the authors' relatively lax standards from study 6.

In Study 6, the lack of temporal stability severely undermines the test-retest reliability of the measure - correlations on their own are not adequate, if scores change overe time, which they do on two of the top three factors.

Why were 4 items dropped at the end of Study 6? As this was the most underpowered study of the two in this section, it seems unclear why one would make that change.

In Study 7, the authors perform yet another CFA. Why? And what criteria are used for concluding that their measure was better than the others? That was not apparent from Table 5. In short, have the authors really just reinvented the wheel here, using a relatively long, psychometrically weak measure to do a job that was already done just as well by existing measures (in the case of Table 6, it looks as if the IES-2 already does this job for the core weight variables).

The norms in Table 7 are meaningless if there is no validation.

6. PLOS authors have the option to publish the peer review history of their article (what does this mean?). If published, this will include your full peer review and any attached files.

Reviewer #1: No

Reviewer #2: No

---

## [Author Response · Author response to Decision Letter 0]

6 Mar 2020

General response

We would like to thank both the reviewers and the academic editor for their helpful comments on our paper. We greatly appreciate the time and effort they spent to review our paper. We have attempted to address all comments in the revised manuscript. In addition, we have embedded our responses within the revision letter text.

Journal Requirements:

->We have now adapted our title, affiliations, contributorship, tables (and values with decimals in text), and file names to meet PLOS ONE’s style requirements.

->We have added the following information in the revised cover letter. The data of Studies 1-7 have been uploaded in the OSF Repository (Creation date: 06/03/2020, URL: https://osf.io/w3guh/?view_only=406839237c9f4bc09ff154d065c17da3). For the purpose of this review, we provide the editors and reviewers with the view-only link to the project. The data will become publicly available once the paper is accepted for publication.

3. Your ethics statement must appear in the Methods section of your manuscript. If your ethics statement is written in any section besides the Methods, please move it to the Methods section and delete it from any other section. Please also ensure that your ethics statement is included in your manuscript, as the ethics section of your online submission will not be published alongside your manuscript.

->Our ethics statement can be found in the Methods section of the paper.

4. Thank you for including your ethics statement: "The studies were conducted according to the guidelines laid down in the Declaration of Helsinki and complied with the code of conduct of Wageningen University. Written consent was obtained for all survey participants.". 

For studies reporting research involving human participants, PLOS ONE requires authors to confirm that this specific study was reviewed and approved by an institutional review board (ethics committee) before the study began. Please provide the specific name of the ethics committee/IRB that approved your study, or explain why you did not seek approval in this case.

->We have uploaded the retrospective approval from the Social Science Ethics Committee of Wageningen University in the submission system and we have adapted our ethics statement in the manuscript (lines 174-178) as well as in the submission form.

 

Reviewers' comments:

Reviewer #1: The authors present the results of an effort to develop a new measure of internally regulated eating. They make a reasonable case for the need for such a measure given extant measures of associated constructs. They mine the literature for dimensions that conceptually define internally regulated eating and develop a self-report measure that maps closely onto a domain-space defined by those dimensions. Their work is among the best I have read in terms of scale development and initial validation. Item generation and selection are thoughtful, and the psychometric analyses are thorough and informative. All in all, it is, in my view, a strong manuscript.

->We thank the reviewer for his nice words. We are happy that he/she acknowledges our hard work.

Here are three minor considerations:

1. It is not clear to my what is to be gained from the series of one-factor models evaluated as described on pp. 12-13. The problem with these analyses is that do not examine the critical question of whether items written for a particular dimension uniquely indicate the intended dimension. It leads to reporting loadings in Table 1 that appear to be from a single model but in fact are from many small models. All I needed to see was the test of the full model, reported around the middle of p. 13, and my preference would be to see the loadings from that model in a supplemental table.

->We agree with the reviewer that this part is confusing and perhaps less interesting for the reader. Thus, we removed it from the text (see edits in lines 161-162, 282-301, 375-377) and we added the requested table in the Supplementary material (S1 Table).

2. I understand the context distinction, but building it into the measures significantly increases the length and complexity of the measure if used as the authors intend. The question is whether information about internally regulated eating from specific contexts offers predictive advantages in hypothesis tests. If, as an example, one looks at the dieters-non-dieters comparisons in Table S1, Cohen's ds are virtually identical for SS and SES and not substantially different for SH and SEH. In terms of the latter two, it is notable that the inconsistent d is for the emotional context, which seems to be operating somewhat differently than the other two here and in other results. Similarly, the stability coefficients in Table 4 are very similar within dimension across context (e.g., .57, .64 and .62 for SH). Scores are collapsed across context within dimension in Tables S4 and S5, making it impossible to determine whether predictive or incremental validity varies as a function of context. In short, I think the authors need to make a stronger case for building context into the measure. My recommendation would be to, if at all possible, drop the distinction.

->We agree with the reviewer that the context distinction makes our scale lengthier and more difficult to use. Thus, it makes sense to examine whether the addition of three contexts has predictive advantages over the neutral counterparts of the SH, SS, SEH, and SES subscales. However, we believe that Cohen’s ds and stability coefficients are not appropriate indicators of the contexts’ predictive advantages. To test this empirically we conducted additional analyses with the data of Study 7. Specifically, we tested whether the full subscales (SH, SS, SEH, and SES), which included all three contexts each, account for a larger amount of variance in each outcome measure compared to the neutral counterpart of each subscale. We addressed this issue in lines 773-789 and we added supplementary tables (S7, S8, S9, S10) and a figure (S2 Fig) to present the relevant results. Since the 45-item and 21-item versions of the scale are comparable we leave it to the user to decide which version to use depending on their specific interests (see also edits in lines 848-854 and S1 Appendix).

3. One issue for measures like this is how they should be scored. For the validity analyses, the authors consider both a total score and a score by dimension (ignoring context). My inference is that they would favor either scoring depending on research question. It is worth noting that researchers sometimes use only a subset of subscales from a measure like this. When the first-order factors are reflective indicators of the general factor, a total score can be meaningfully interpreted. In the case of formative indicators, as the authors have cast the first-order factors, that is not the case. The general factor is not fully defined unless all of the formative indicators are included in a composite.

->We thank the reviewer for raising this important issue about the use of our scale. We now clarify this in lines 840-848. 

A very minor matter: On p. 33, just before the general discussion, the authors suggest differences that imply comparisons of correlation coefficients, which they did not do. "Outperformed" implies a significantly larger coefficient, which requires a test comparing them. 

->We changed the phrasing in lines 768-770.

One somewhat related and even more minor matter: Please include the actual coefficients for all pairwise rs in Table 5. The use of "NS" in a table like this is a meta-analyst's nightmare and, generally speaking, unnecessarily hides information that need not be hidden.

->We replaced the “NS” with the actual values as requested. We also corrected the values in the third column (correlations of IES-2) because we realized that the ones we had included in the original version were from an adapted version of IES-2 (in which all reverse-scored items had been excluded).

These matters are very minor and do not detract from my overall positive view of the manuscript. I congratulate the authors on a fine effort.

->Once again, we thank the reviewer for his particularly useful comments that have helped us to improve our paper and scale substantially.

 

Reviewer #2: The authors report on the development of a new measure (MIRES). My first impression is that this is an extremely long paper, for its content. As such, it loses focus and clarity about its aims. By the end, I was not clear what psychological or clinical issue the authors were addressing, or what the implications of the work would be.

->We thank the reviewer for the time spent on reviewing our paper and for the detailed comments. In this paper we present a large number of studies and analyses that we conducted for the thorough development and validation of the MIRES. We believe that all the information we provide is necessary and useful for the users of the scale. This prevents us from reducing the length of the paper substantially. However, we have made several efforts (already in the original version of the paper) to help the readers keep up with paper and its aims. Specifically, we use the following structure. First, we introduce the aim of the paper early on, in lines 60-73. Second, under the Methods section we present the overview of studies with their specific aims (lines 154-171), accompanied by Figure 1 (line 180), which is a graphical summary of the whole paper. Third, the results of each of the main studies are summarized in the discussion section of each study, followed by an overall wrap up in lines 801-823 of the General discussion section. Finally, we made some edits in lines 792-800 to remind readers of the main construct that we examine in this research.

More importantly, the presentation failed to meet a number of basic psychometric requirements in developing such a measure.

->We regret to see that the reviewer is not satisfied with the amount of evidence we present on the scale’s psychometric properties. We believe this has to do with the way Studies 1-4 had been presented in the initial version of the manuscript. As we discuss later on, we now provide additional information on this preliminary work.

MAJOR ISSUES

There is a core problem with the reporting of the initial samples used in studies 3-7. The ages of the first two samples are substantially different, making the findings hard to compare. Without age-based norms for the original measure, comparability of scores or of factor structure is not safe to assume.

->Starting with college samples is a common practice in scale development given the low costs and convenience of collecting such data. While many scale developers often settle with such data, we chose to go one step forward and test the properties of our scale in large community samples to increase the external validity of our research. We do not think that comparison of findings between college and community samples is problematic because in our studies we are mainly interested in the covariance between items not the variance within items. While means and standard deviations may vary depending on the sample, the covariance between items is not expected to do so (or at least not substantially).

Furthermore, the reason why we do not provide age-based or gender-based norms is because if we were to do this, we would first have to test our scale for measurement (in)variance across age groups or sexes. Since this would increase the length of the paper even more, we decided to leave this out for now and maybe present these data in another publication.

Studies 1-4 are simply not reported in anything like adequate detail. Where are the initial factor analyses that I presume must have been conducted? What factors emerged, using what methods? Was there replication? Internal consistency? Test-retest reliability? Most importantly, how do the authors justify a sample in studies 3 and 4 that are simply too small to allow for reliable factor derivation (80 items require a sample of 800 participants PER STUDY to ensure reliable outcomes)? In short, the derivation of a pool of 49 items is just not safe. Nor do we have any reason to assume that the factors that are addressed in studies 5-7 are in any way related to what was found in Studies 1-4.

->We agree with the reviewer that these studies had been presented very briefly in the original paper. To enrich this section (see edits in lines 204-232), we added information on the rationale of dropping items, the sample size justification, and the most important findings of these studies, which helped us to bring the model to its current form.

The main reason why we do not present any hard data on these studies is because this preliminary work took place during the initial steps of developing our conceptual model. In the course of these studies our model was adjusted several times since we came across empirical findings we had not initially expected. Thus, should we want to add statistics on these studies we would have to present a long course of model adaptations and this would make the paper unnecessarily lengthier and more difficult to read. It would be challenging to keep the paper in its current concise form with a clear conceptual model to begin with. After all, in studies 5-7, which are the large-scale studies we conducted, we tested all important psychometric properties that are commonly reported in scale development (or even more). Thus, we believe that readers and users have all evidence needed to judge the psychometric quality of the scale.

For Study 5, the construct validity element is not clear. What were the t-values per measure, and what was the P value adopted? Why not use binomial regression to determine the key MIRES variables? And why use one simple measure like this (dieting/not dieting), when understanding such a complex construct? An appropriate index would have been to determine whether the MIRES is a better predictor than other measures of the same constructs, to show that the MIRES is a more clinically useful measure.

->We added the t values in Table S2 as requested (we also found some small inaccuracies in the Cohen’s d column and we corrected them). As we now more clearly explain in lines 364-368, in Study 5 we tried to get preliminary evidence on the scale’s construct validity. Our aim was not to make an accurate prediction of dieting behaviour, but to show that non-dieters score higher in all MIRES subscales than dieters. This supports the very nature of the constructs we investigate in a broad sense, since internally regulated eating is by nature a non-diet eating style. Also, it was not the aim of this study to test the scale for criterion or incremental validity, as is suggested by the reviewer. Evidence on those types of validity are presented in Study 7 (Table S6 and Table 6). 

If Study 6 really did use the same analyses as Study 5, where is the report of the CFA? If one was conducted, then the sample in Study 6 appears to be underpowered, even by the authors' relatively lax standards from study 6.

->In lines 401-415 we describe the exact procedure that we followed to analyse the data of Study 6. We now clarify that we conducted individual analyses per subscale. Because we ran and compared up to seven CFA models per subscale (19 subscales), we decided not to report these results in a table, but rather to describe them briefly in lines 416-425, accompanied by Table 4. Since analysis was conducted per subscale the sample size was adequate for testing all first-order factor models. For testing the stability of second order factors the sample was indeed a bit small (4:1 ratio). We addressed this issue in lines 391-394.

In Study 6, the lack of temporal stability severely undermines the test-retest reliability of the measure - correlations on their own are not adequate, if scores change over time, which they do on two of the top three factors.

->The reviewer is right that the scores of two subscales (IT and FL) change over time despite the fact that stability coefficients are adequate. This means that there is variation over time across the whole studied population. The change in means across time is a stricter test of stability than correlations and is not always reported by scale developers. However, we deem this important and so we have reported these results, hoping that readers will appreciate it. We made some edits in lines 456-460 and 813-815 in order to address this issue more explicitly.

Why were 4 items dropped at the end of Study 6? As this was the most underpowered study of the two in this section, it seems unclear why one would make that change.

->As we now more explicitly explain in lines 435-446, we decided to drop four items in order to reduce the length of the scale and to have the same number of items per subscale. The reason why we dropped those items after the end of Study 6 was because we wanted to test the stability of the MIRES subscales first. The deletion of items was not based on findings from Study 6 but on the meaning of items, because all subscales were initially found to be stable on the basis of stability coefficients, which were adequate. The change in means was an extra analysis that we conducted at a later stage when writing the paper. By that time all studies had been completed.

In Study 7, the authors perform yet another CFA. Why? And what criteria are used for concluding that their measure was better than the others? That was not apparent from Table 5. In short, have the authors really just reinvented the wheel here, using a relatively long, psychometrically weak measure to do a job that was already done just as well by existing measures (in the case of Table 6, it looks as if the IES-2 already does this job for the core weight variables).

->As we explain in lines 470-473, in Study 7 we test for the first time the full formative model of MIRES, including the six reflective items. 

The criteria we used to decide whether our scale is better than the others included 1. the ability to predict outcome measures – Table S6 and Table 5 (descriptive comparison of scales) and 2. the incremental variance accounted for by MIRES – Table 6 (statistical testing). As can be clearly seen in Table 6, our scale accounts for significant amounts of extra variance in most outcome measures above and beyond the variance accounted for by IES-2 and ecSI-2 (see also edits in lines 715-720, which are intended to make this point clearer). Exceptions are the weight outcomes for which all three measures are relatively weak predictors.

The norms in Table 7 are meaningless if there is no validation.

->Perhaps the reviewer has a different interpretation of norm scores than we do. We do not use norm scores in relation to predictive validity, i.e., expecting that individuals from different norm score categories would behave differently or show different outcomes, which would indeed require further validation. We use them from a descriptive point of view, to allow researchers to use them as reference and to compare between different groups or populations. Also, a subject’s raw score has little meaning, unless we know the subject’s position relative to others in some group/population. Thus, our norm scores provide the basis for comparing different groups or an individual with a group. We now clarify this in lines 742-745. As we explained earlier, we do not provide age-based or gender-based norms because this would require additional analyses that could perhaps be included in another publication. Finally, we re-calculated the norms based only on the data from the US sample because it is more useful to have country-based norms (see edits in lines 736-738 and Table 7).

---

## [Decision Letter · Decision Letter 1]

28 Apr 2020

PONE-D-19-26656R1

Development and validation of the multidimensional internally regulated eating scale (MIRES)

PLOS ONE

Dear Mrs Palascha,

Thank you for submitting your manuscript to PLOS ONE. After careful consideration, we feel that it has merit but does not fully meet PLOS ONE’s publication criteria as it currently stands. Therefore, we invite you to submit a revised version of the manuscript that addresses the points raised during the review process.

We would appreciate receiving your revised manuscript by Jun 12 2020 11:59PM. To enhance the reproducibility of your results, we recommend that if applicable you deposit your laboratory protocols in protocols.io, where a protocol can be assigned its own identifier (DOI) such that it can be cited independently in the future. For instructions see: http://journals.plos.org/plosone/s/submission-guidelines#loc-laboratory-protocols

We look forward to receiving your revised manuscript.

Kind regards,

Gian Mauro Manzoni, Ph.D., Psy.D.

Academic Editor

PLOS ONE

Reviewers' comments:

Reviewer's Responses to Questions

**Comments to the Author**

1. If the authors have adequately addressed your comments raised in a previous round of review and you feel that this manuscript is now acceptable for publication, you may indicate that here to bypass the “Comments to the Author” section, enter your conflict of interest statement in the “Confidential to Editor” section, and submit your "Accept" recommendation.

Reviewer #1: All comments have been addressed

Reviewer #2: (No Response)

2. Is the manuscript technically sound, and do the data support the conclusions?

Reviewer #1: (No Response)

Reviewer #2: No

3. Has the statistical analysis been performed appropriately and rigorously? 

Reviewer #1: (No Response)

Reviewer #2: No

4. Have the authors made all data underlying the findings in their manuscript fully available?

Reviewer #1: (No Response)

Reviewer #2: Yes

5. Is the manuscript presented in an intelligible fashion and written in standard English?

Reviewer #1: (No Response)

Reviewer #2: No

6. Review Comments to the Author

Reviewer #1: (No Response)

Reviewer #2: I do not understand the authors' conclusion that, having spent all that effort to demonstrate that the short and long version of the paper are equivalent, it should be up to the reader to decide which version to use. They present no rationale for having two versions, and I would recommend that they should focus on the shorter version, as that will eventually take a lot less time on the part of patients, clinicians and researchers.

The authors state that they clarity their preferred scoring in lines 840-848, but they do not.

The paper remains excessively long, and the authors do not justify that length in their response letter. I did not find that the very minor changes alluded to made it any clearer what the aims, clinical issue or implications were.

The lack of norms is defended as potentially making the paper longer (which would be more than offset if the authors had actually shortened the paper in response to previous feedback). However, the authors ignore the key issue that was identified - the lack of comparability between samples.

The authors decline to provide key findings in Studies 1-4. The same applies later to other studies. That means that they are not replicable. That is a fundamental error in scientific communication, and I could not support publication of any such work, including this paper.

The authors do not justify the lack of validation for their scores in Table 7, and they remain meaningless as a result.

---

## [Author Response · Author response to Decision Letter 1]

11 Jun 2020

General response

We want to thank the reviewers for taking the time to review our paper for a second time. We are happy that one of the reviewers was satisfied with the first revision and gave consent for acceptance of our paper. We equally appreciate the effort of the second reviewer to provide us with additional insights for improving our paper. We found some comments challenging, but we feel we have managed to address them all. The paper has been substantially revised in response to the reviewer comments. Most important changes are: (i) providing the details on studies 1-4, which we originally addressed more as pre-studies, but have now become an integral part of the “full history” of the development of the 21-item MIRES scale, (ii) strong recognition of the short version of the scale (21 items instead of 45 items), and (iii) shortening of the paper (from 12375 words to 11953 words) despite adding the requested material on studies 1-4, primarily obtained in the background information to the scale. We are grateful for the reviewer’s critical look at the paper and feel that the paper has benefited greatly from the reviewer’s comments. Please find our specific responses embedded in the text below.

Reviewers' comments:

Reviewer #2: 

1. I do not understand the authors' conclusion that, having spent all that effort to demonstrate that the short and long version of the paper are equivalent, it should be up to the reader to decide which version to use. They present no rationale for having two versions, and I would recommend that they should focus on the shorter version, as that will eventually take a lot less time on the part of patients, clinicians and researchers.

In hindsight, we agree with the reviewer that most potential users may prefer to use the short version of the scale and that they should be able to access it quickly. In the revised manuscript, we now appreciate this from the beginning by including the following changes: 

• We now mention the short version of the scale in the abstract (lines 30-33) and introduction (lines 92-95) so that readers expect this as the final outcome.

• We have made edits in lines 1031-1035 to contrast the two versions and show the advantage of the short version more explicitly.

• We moved Fig S2, which illustrates the simplified version, into the paper (see Fig 2 in lines 1037-1040) and moved Fig 2, which illustrates the full version, to the supplementary material (see S2 Fig). In the new Fig 2, we replaced the item codes with the actual items so that readers can see directly how the scale looks. S1 Appendix still provides information on how the full version was administered in our studies (to be used as a reference to the version we used in our studies) (lines 525-538). Although we are now directing users to the short version, we believe it is important to present the full version for transparency purposes.

• We explicitly advise potential users to go for the short version of the scale (lines 1107-1109).

• We skipped lines 1125-1130.

2. The authors state that they clarity their preferred scoring in lines 840-848, but they do not.

The main point raised in the first review was that users should not measure only a subset of subscales if they want to make conclusions about IRE style because all subscales of a formative construct need to be measured for the formative construct to be measured. This is the issue we addressed in lines 840-848 of the first revision. In the revised manuscript, we now explicitly refer to the appropriate scoring depending on the case (lines 1114-1119).

3. The paper remains excessively long, and the authors do not justify that length in their response letter. I did not find that the very minor changes alluded to made it any clearer what the aims, clinical issue or implications were.

We agree with the reviewer that the paper is long. We made additional efforts to shorten the paper. As we mentioned in the first revision, we believe all information about the scale’s properties is necessary and useful for the scale’s users; therefore, we tried to cut down parts of the paper that are less critical, i.e., information that can be traced back to existing literature. Effectively, we managed to reduce the length by 422 words and achieved that through the following changes:

• We removed examples of items and evidence on validity for the various measures we used in Study 7 to validate our scale (lines 754-881).

• We shortened the conceptual part by skipping background information and evidence of adaptivity for the IRE characteristics (lines 134-180), because a detailed overview of the conceptual model can be found in our theoretical paper that is freely accessible online. We now present only the definitions of the IRE characteristics and we explain how we operationalize them. In lines 126-128 we refer readers to the complete overview of the model.

• We skipped the description of the temporal stability assessment process because readers can find the full description of the method in the paper of Steenkamp & Van Trijp (lines 659-668).

• We skipped the section on norms (lines 982-1001); but see also point 4 below.

• We also made smaller edits throughout the paper to aid brevity and clarity (e.g., see lines 57-92).

• We skipped the wrap up of findings in the general discussion (lines 1059-1081) and adapted the abstract to contain all this information.

4. The lack of norms is defended as potentially making the paper longer (which would be more than offset if the authors had actually shortened the paper in response to previous feedback). However, the authors ignore the key issue that was identified - the lack of comparability between samples.

Even though we are not entirely sure, we believe that the reviewer aims to communicate that the whole scale development process should be conducted with comparable samples and that should non-comparable samples be used, measurement invariance of the model would have to be tested to ensure that the model holds its properties across the different samples. 

We agree that measurement invariance testing is important when new scales are developed and perhaps should be conducted not only for different age groups but also for gender groups or other clinically relevant groups such as people with obesity. We now address the issue of comparability between samples in lines 1092-1105 and we have conducted additional analyses to provide preliminary evidence on the measurement invariance of our model across the age groups of interest (18-34 and 35-65). We realized that the sample sizes in our subgroups (in both studies 5 and 7) are too small to conduct typical measurement invariance testing (at least 980 participants per group are required to test the model), thus, we tried to find an alternative solution. Instead of testing for configural, metric, scalar, and residual invariance in a stepwise process as is usually done, we tried to test our model for full invariance at once by fixing all parameters to the values obtained from the full sample of study 7 and then fitting this model in each of the subgroups (18-34 and 35-65). In this way, the model had less parameters to be estimated and the sample sizes of our subgroups were adequate to provide reliable estimates. To warrant model identification we left only seven parameters free, the regression coefficients of the seven formative indicators. The fit of the model was good in both age groups providing preliminary evidence for the measurement invariance of our model. We further discuss this issue in the directions for future research (lines 1150-1153), where we also make a link to the development of norms.

5. The authors decline to provide key findings in Studies 1-4. The same applies later to other studies. That means that they are not replicable. That is a fundamental error in scientific communication, and I could not support publication of any such work, including this paper.

Per the reviewer’s request we have now included extensive information on these early studies as part of the “full history” of the MIRES development, both conceptually and in interaction with measurement. Studies 1-4 were initially considered pilot studies leading to the 52 initial items of the MIRES scale. These studies have helped us to develop that initial item set based on conceptual and empirical considerations. We absolutely agree with the reviewer on the importance of transparency and reproducibility of research. In the previous version we had focussed on the later stages and outcome of MIRES, and we now shift to describe the full history of the process with a stronger focus also on the early stages of scale development.

To allow readers to understand the full process of scale development that we followed, we added all details of Studies 1-4 in the paper and we made all data of this project freely accessible to the public (see edits in lines 30-37, 98-114, 206-215, 309-508). This will allow other researchers to replicate our work and to build further on it. Taking into account that there is always a degree of subjectivity in the scale development process (e.g., when selecting items to retain or to drop) we believe that readers are now fully equipped to understand the decisions we took during the scale’s trajectory by accessing the full data of the project.

6. The authors do not justify the lack of validation for their scores in Table 7, and they remain meaningless as a result.

The reviewer is right that descriptive kind of norms are less informative compared to validated norms that have been developed in relation to important clinical outcomes. After all, norms should develop from multiple studies and not from a single study. For these reasons and to help reduce the length of the paper we decided to skip the section on norms (lines 982-1001) and to discuss this issue in the discussion section as a promising direction for future research (lines 1152-1153).

---

## [Editor Report · Decision Letter 2]

24 Jul 2020

PONE-D-19-26656R2

Development and validation of the multidimensional internally regulated eating scale (MIRES)

PLOS ONE

Dear Dr. Palascha,

Thank you for submitting your manuscript to PLOS ONE. After careful consideration, we feel that it has merit but does not fully meet PLOS ONE’s publication criteria as it currently stands. Therefore, we invite you to submit a revised version of the manuscript that addresses the points raised during the review process.

First of all, allow me to compliment you on your remarkable research, which is undoubtedly a contribution to the assessment of eating behaviors. I have reviewed your paper completely and the responses to the reviewers and I think you have done a good job. However, there is one comment on which I agree with reviewer 2 and that is the one referring to the length of the manuscript, although, more than the length, I have my misgivings about the structure.

In particular, I consider that all the studies are, in fact, part of one same study (with different samples and analyses), whose division into studies makes the manuscript somewhat complicated. I believe that there are parts of the report that are sufficient to be mentioned within the procedure (studies 1 to 4), putting the different pools of items as supplementary material, while, for the remaining studies, it would only be necessary to specify the different samples and the different analyses.

If you do not agree with modifying and unifying the structure of the study, please try to make it as integrated and simple as possible. I know that, at this stage of the work, it can be tedious to make this type of cosmetic transformation, however, it is necessary for me to make this recommendation since, in order for your work to have the scope it deserves, it is not only necessary to have the technical rigour (which your work already has), but it also needs to be friendly and easy to understand for the reader.

We look forward to receiving your revised manuscript.

Kind regards,

Rodrigo Ferrer, Ph.D.

Academic Editor

PLOS ONE

Additional Editor Comments (if provided):

Dear Authors:

First of all, allow me to compliment you on your remarkable research, which is undoubtedly a contribution to the assessment of eating behaviors. I have reviewed your paper completely and the responses to the reviewers and I think you have done a good job. However, there is one comment on which I agree with reviewer 2 and that is the one referring to the length of the manuscript, although, more than the length, I have my misgivings about the structure.

In particular, I consider that all the studies are, in fact, part of one same study (with different samples and analyses), whose division into studies makes the manuscript somewhat complicated. I believe that there are parts of the report that are sufficient to be mentioned within the procedure (studies 1 to 4), putting the different pools of items as supplementary material, while, for the remaining studies, it would only be necessary to specify the different samples and the different analyses.

If you do not agree with modifying and unifying the structure of the study, please try to make it as integrated and simple as possible. I know that, at this stage of the work, it can be tedious to make this type of cosmetic transformation, however, it is necessary for me to make this recommendation since, in order for your work to have the scope it deserves, it is not only necessary to have the technical rigour (which your work already has), but it also needs to be friendly and easy to understand for the reader.
---

## [Author Response · Author response to Decision Letter 2]

9 Sep 2020

Response to reviewers

Additional Editor Comments (if provided):

Dear Authors:

First of all, allow me to compliment you on your remarkable research, which is undoubtedly a contribution to the assessment of eating behaviors. I have reviewed your paper completely and the responses to the reviewers and I think you have done a good job. However, there is one comment on which I agree with reviewer 2 and that is the one referring to the length of the manuscript, although, more than the length, I have my misgivings about the structure.

In particular, I consider that all the studies are, in fact, part of one same study (with different samples and analyses), whose division into studies makes the manuscript somewhat complicated. I believe that there are parts of the report that are sufficient to be mentioned within the procedure (studies 1 to 4), putting the different pools of items as supplementary material, while, for the remaining studies, it would only be necessary to specify the different samples and the different analyses.

If you do not agree with modifying and unifying the structure of the study, please try to make it as integrated and simple as possible. I know that, at this stage of the work, it can be tedious to make this type of cosmetic transformation, however, it is necessary for me to make this recommendation since, in order for your work to have the scope it deserves, it is not only necessary to have the technical rigour (which your work already has), but it also needs to be friendly and easy to understand for the reader.

Response:

We would like to thank the editor for the time he spent on reading our original paper and the subsequent revisions and for his particularly kind and gratifying words regarding our work. We found the additional suggestions of the editor very useful and we believe the new edits we made have made the paper more “to the point” and easier to read. In line with the editor’s suggestion, we now present studies 1-7 as different parts of a single study with a unified Methods section and a unified Analysis and Results section. We refer to studies 1-4 as the preparatory work of this research (mentioned briefly under the Methods section) that led to the structure we used in large-scale testing of the MIRES. The data of all studies are still available online in the OSF repository (we changed the study numbers to descriptive titles so that readers can link more easily to our paper) and the versions of the MIRES items are now presented within a single document in the supplementary material (S2 Appendix). We also adapted Figure 1 so that it doesn’t refer specifically to studies and doesn’t include the sample sizes but is more descriptive of the properties of MIRES that we tested in this research. Finally, we have added a large paragraph in lines 948-985 of the Discussion section to summarize the main findings of this research because the individual discussions sections we used in the previous version have now been deleted. With these changes we managed to reduce the length of the paper by 2,619 words.

As an additional point, we now cite the latest version of our theoretical paper, which is also at the second round of revision in another journal (Palascha A, van Kleef E, de Vet E, van Trijp H. Internally Regulated Eating Style: A Comprehensive Theoretical Framework. OSF [Preprint, version 2]. 2019 [cited 2020 Sep 4]. Available from: https://osf.io/rmbft/ doi: 10.31219/osf.io/rmbft).

We hope these changes to have produced a paper that meets the criteria for publication at PlosOne, but we are happy to make more changes if the editor deems this necessary. We are looking forward to the editor’s response.

---

## [Editor Report · Decision Letter 3]

16 Sep 2020

Development and validation of the multidimensional internally regulated eating scale (MIRES)

PONE-D-19-26656R3

Dear Dr. Palascha,

We’re pleased to inform you that your manuscript has been judged scientifically suitable for publication and will be formally accepted for publication once it meets all outstanding technical requirements.

Kind regards,

Rodrigo Ferrer, Ph.D.

Academic Editor

PLOS ONE
---

## [Editor Report · Acceptance letter]

30 Sep 2020

PONE-D-19-26656R3 

Development and validation of the multidimensional internally regulated eating scale (MIRES) 

Dear Dr. Palascha:

I'm pleased to inform you that your manuscript has been deemed suitable for publication in PLOS ONE. Congratulations! Your manuscript is now with our production department. 

Kind regards, 

on behalf of

Dr. Rodrigo Ferrer 

Academic Editor

PLOS ONE